# How to Train Your Advisor:
# Steering Black-Box LLMs with ADVISOR MODELS

**Parth Asawa** [* 1]  **Alan Zhu** [* 1]  **Abigail O'Neill** [1]
**Matei Zaharia** [1]  **Alexandros G. Dimakis** [1 2]  **Joseph E. Gonzalez** [1]

## Abstract

Frontier language models are deployed as black-box services, where model weights cannot be modified and customization is limited to prompting. We introduce ADVISOR MODELS, a method to train small open-weight models to generate dynamic, per-instance natural language advice that improves the capabilities of black-box frontier models. ADVISOR MODELS improve GPT-5.2's performance on RuleArena (Taxes) by 27.4%, reduce Gemini 3 Pro's steps taken in SWE agent tasks by 24.6%, and outperform static prompt optimizers in personalizing GPT-5 to user preferences (85-100% vs. 40-60%). We also find that advisors are transferable: an advisor trained with a low-cost student model still transfers improvements to a frontier model. Moreover, ADVISOR MODELS are robust: we observe no degradation on other benchmarks than the pipeline is trained on. Our method shows how to perform parametric optimization for black-box frontier models in a practical and cost-effective way.

## 1. Introduction

Can you specialize a model when you don't have it's weights? Frontier models like GPT-5 and Claude 4.5 (OpenAI, 2025b; Anthropic, 2025b) are deployed as black-box API services where weights are inaccessible, making traditional fine-tuning impossible. On the other hand, static prompts can't adapt to diverse inputs and are constrained by context-length limitations. These limitations are especially acute when applications need deep specialization (e.g., low-resource languages, complex rule-following and com-

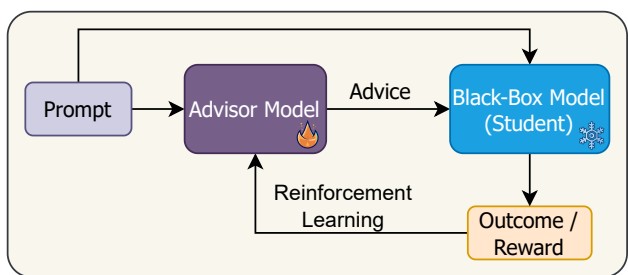

*Figure 1.* **ADVISOR MODELS combine open-source models with black box models.** ADVISOR MODELS trains an advisor to generate instance-specific advice that is injected in-context to steer a frozen black-box model. Rewards from the environment of the final output are used to train the advisor with reinforcement learning.

pliance, SWE agent efficiency in a particular repository) or personalization (heterogeneous user preferences).

We introduce **ADVISOR MODELS** (Figure 1), a method for training a lightweight model to reactively steer a black-box model by generating dynamic input-specific advice. Crucially, ADVISOR MODELS can be parametrically optimized with reinforcement learning, using task-specific rewards derived from the student model's final outputs. This process enables the compound system to learn and adapt without access to the student model's parameters or gradients, making our approach effective for steering or personalizing black-box models. The *learning to advise* formulation re-frames prompting from a static search problem into learning a policy that generates custom advice for every instance.

Across a range of tasks, we show ADVISOR MODELS (1) *lift frontier model performance* on reasoning tasks where domain knowledge is sparse in pretraining data, and (2) *learn hidden preferences* for personalization where preferences vary per-instance in ways that cannot be known *a priori*. We further show that gains from ADVISOR MODELS exceed those of current static prompt optimizers. Finally we show that advisors are transferable: advisors trained with a low-cost student model would still transfer improvements to a frontier student model, resulting in low overall training cost.

**Improving Frontier Models.** In Section 4, we show that ADVISOR MODELS trained on low-cost black-box models (e.g., GPT-4o mini, Gemini 2.5 Flash) are effective at im-

---
*Equal contribution. Code available at: https://github.com/az1326/advisor-models. [1]University of California, Berkeley [2]Bespoke Labs. Correspondence to: Parth Asawa <pgasawa@berkeley.edu>, Alan Zhu <aczhu@berkeley.edu>.

*Proceedings of the 43rd International Conference on Machine Learning*, Seoul, South Korea. PMLR 306, 2026. Copyright 2026 by the author(s).

proving frontier models (e.g., GPT-5, Gemini 3 Pro) on tasks spanning specialized reasoning and personalization. On specialized reasoning tasks, advisors improved GPT-5's accuracy on a tax filing benchmark from 67.2% to 85.6% and reduced Gemini 3 Pro's efficiency on SWE agent tasks from 31.7 average steps to 26.3 while retaining resolve rate. On personalization tasks, advisors succeeded at personalizing GPT-5's outputs, improving reward on the tasks from 40-60% reward to 85%+. In stark contrast, static prompt optimization methods like the state-of-the-art GEPA (Agrawal et al., 2026) and Profile-Augmented Generation (Richardson et al., 2023) don't improve models on these tasks.

**Transferability.** Despite evaluating with student models that are different and stronger than those used during training, ADVISOR MODELS learn to generate advice that continued to provide substantial gains even after transferring to frontier models. Additionally, advisors trained on one model family (e.g., GPT) can be transferred to another (e.g., Claude), further demonstrating the generalizability of advisors (Appendix D.1).

**Robustness.** As the ADVISOR MODELS architecture enables parametric learning without modifying the student model, we reduce the risk of unintended consequences such as catastrophic forgetting. We verify this resilience by showing ADVISOR MODELS pipelines maintain performance in one benchmark when trained for another.

In this paper, we demonstrate the potential of ADVISOR MODELS as a new paradigm for optimizing black-box systems in an interpretable way. Our contributions are:

1. We introduce **ADVISOR MODELS**, a novel method for dynamically steering frozen, black-box models using trainable policies.

2. We demonstrate its effectiveness in settings including specialized reasoning and personalization, where it can **improve the performance of frontier models** such as GPT-5 and significantly outperforms static baselines.

3. We demonstrate the ability to **transfer advisors** across black-box models and show that the framework leads to **robustness** on untrained tasks even with a specialized model.

## 2. Related Work

**Static Prompt Optimization.** Some prior work focuses on automatically discovering fixed prompts for black-box models, using gradient-free search, evolutionary algorithms, or reinforcement learning to improve task performance across datasets (Khattab et al., 2022; 2024; Opsahl-Ong et al., 2024; Zhou et al., 2023; Yang et al., 2024a; Agarwal et al., 2025; Fernando et al., 2024; Agrawal et al., 2026). While effective

in some settings, such methods typically yield a single static prompt that must be reused for all instances of a task. These methods are especially limited in agentic settings where the presence of tool output and user feedback may benefit from additional conditioning.

In contrast, our work departs from this paradigm by training a lightweight advisor to reactively produce context-specific natural language advice, leveraging the greater capacity in model weights compared to static natural language. This advice is then injected in-context into the black-box model, allowing adaptation to individual inputs rather than committing to one static prompt.

**Trained Prompt Generators.** A number of related works do train a lightweight model to modify the prompt to a black-box model. Black-Box Prompt Optimization (Cheng et al., 2024) uses training data from an expert that modifies prompts to align with preference labels to train a prompt rewriter. In contrast, ADVISOR MODELS generalize beyond personalization tasks and trains the advisor without needing expert rewrites. PRewrite (Kong et al., 2024) uses RL to train a model that rewrites prompts, but only generates a single prompt for the entire class of problems whereas ADVISOR MODELS generate per-instance advice. IDPG (Wu et al., 2022) does take an instance-specific approach, yet focuses on classification tasks and utilizes soft prompts that are uninterpretable in contrast to the ADVISOR MODELS approach. Both Directional Stimulus Prompting (Li et al., 2023) and Matryoshka Pilot (Li et al., 2025) require an expert labeler to initialize the lightweight model through supervised fine-tuning and constrain the lightweight model's outputs depending on the task (e.g., must output list of keywords for summarization tasks). ADVISOR MODELS assumes no access to an expert labeler and places no constraints on the format of the advice generated.

## 3. ADVISOR MODELS

We introduce ADVISOR MODELS, a method to train a lightweight policy to steer black-box foundation models through natural language. In the most basic setup, the advisor sits between the user input and the black-box model, generating instance-specific guidance that is injected in-context for the black-box model (Figure 1). As open-source models may also serve as the advised model, in this work we will use "black-box" and "student" model interchangeably.

**Design Variants.** The design space for ADVISOR MODELS learning dynamics is broad. For example, building off the 2-step pipeline in Figure 1, we implement a 3-step variant of ADVISOR MODELS. Under this variant, the advisor is given an initial attempt by the student (step 1) to generate its advice (step 2), which the student uses to generate a final revised response (step 3). By starting from an initial student

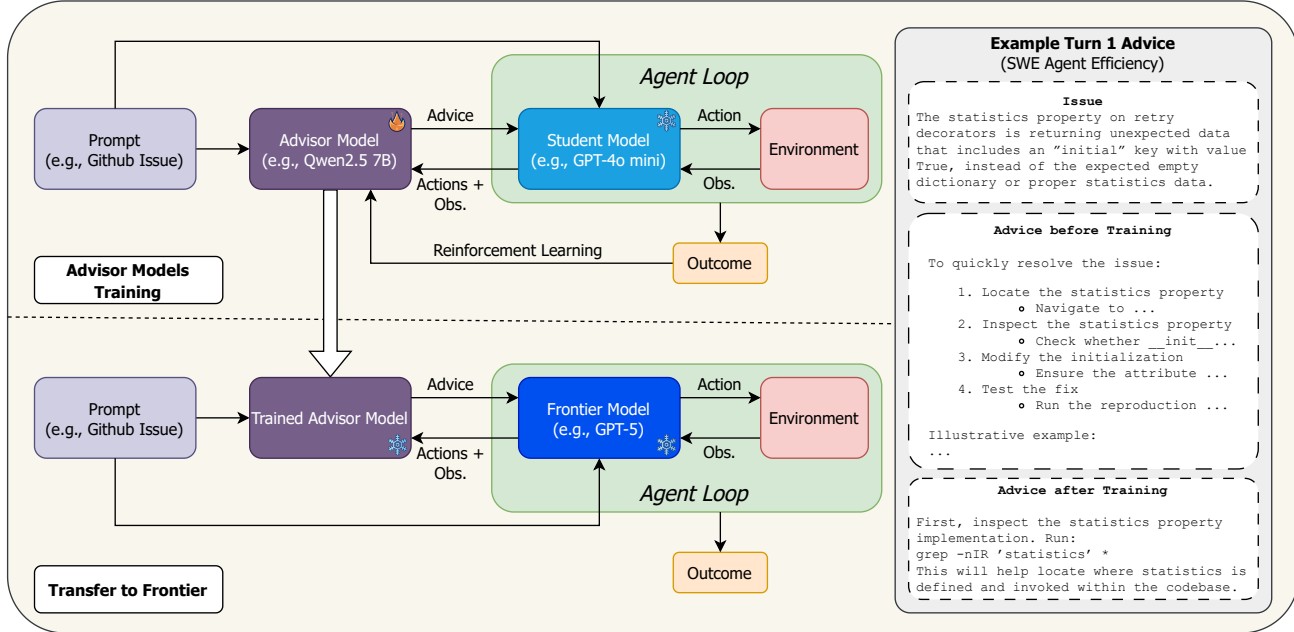

*Figure 2.* **Example of ADVISOR MODELS training and transfer for multi-turn tasks.** The input task is given first to the advisor model to generate turn 1 advice. The input task and the advice are then given to the student agent to guide the student's interactions with the environment. Periodically, information from the student interactions is given to the advisor to generate follow-up advice, which is also provided to the student. The process repeats until a final outcome is reached. The advisor is trained via RL on the reward of the outcome. Advisors can be trained with a small student model and transferred for evaluation with frontier student models without further tuning. Crucially, advisor model RL training can be done with only API access to the black-box model.

generation, we focus the advisor's generations and guard against the advisor providing harmful advice, simplifying the advisor's role to that of a verifier. We found that this setup led to better outcomes in complex tasks.

Figure 2 (top left) illustrates the architecture of ADVISOR MODELS for multi-turn settings with environmental observations, such as SWE agent. Here, the advisor's generation is simply appended to any information usually provided to the student. The student's action(s) and observations are then given back to the advisor to generate the next turn of advice. The student can return after every action, or at regular intervals to save cost (i.e., a single step from the advisor's perspective consists of multiple student actions). Alternatively, the student may choose when to next query the advisor via a tool call, allowing for dynamic number of student steps between advisor turns. Further discussion is provided in Appendix B.1.

**Training.** Training proceeds via reinforcement learning: the advisor samples candidate advice, the black-box model produces outputs conditioned on this advice, and a task-specific reward is computed from the output. We use Group Relative Policy Optimization (GRPO; Shao et al. (2024)) to update the advisor based only on observed rewards. Integrating ADVISOR MODELS into any RL training framework or algorithm is as simple as modifying a step in the environment to make a generation call to the black-box model with the advisor's output (policy's action) before evaluating the

reward on the final student output.

The ADVISOR MODELS design turns prompt engineering into an RL problem, where the advisor learns which advice reliably elicits better performance. Unlike static prompt optimization, the advisor adapts the black-box model on a per-instance basis. No gradient access is needed from the black-box model, allowing for the full leveraging of frontier API-based models. An example of advice given by the advisor prior to and after training is given in Figure 2 (right). Before training, the advisor gives vague high-level advice, but after training the advice becomes clear and actionable.

**Advisor Transferability.** An additional advantage of the modularity of ADVISOR MODELS is the transferability of trained advisors. As advisors learn to generate advice in natural language, they can advise students beyond the model used during training. This may be used, for example, to train an advisor with low-cost student models and transfer to a frontier model student, enabling cost-efficient optimization. An example of transferring an advisor trained on GPT-4o mini to GPT-5 is given in Figure 2 (left).

The benefits of ADVISOR MODELS fall under two major axes:

1. **Reactive, transferable, instance-specific optimization.** A learned policy generates natural language advice conditioned to each input, in contrast to static prompting methods that provide a single fixed context.

Moreover, the learned policy can be used to improve models beyond those used during training.

2. **Leveraging and improving black-box frontier capabilities without modification.** The advisor is updated in gradient space while the black-box model remains unchanged. This produces a robust system that we empirically show improves black-box capabilities while preserving capabilities on untrained tasks, avoiding post-training risks such as catastrophic forgetting.

## 4. Evaluation

We evaluate ADVISOR MODELS on domains with two desiderata: (1) the ability to leverage the strength of black-box models (e.g., reasoning, creative writing), and (2) the presence of domain knowledge not known *a priori* by the black-box model.

Through our experiments, we show that ADVISOR MODELS lifts the capabilities of frontier black-box models (e.g., GPT-5, Gemini 3 Pro) at specialized reasoning tasks and effectively personalizes frontier models at personalization tasks, outperforming other optimizers.

In this section, we train ADVISOR MODELS with small open-weight advisor models (e.g., Qwen2.5 7B Instruct) and low-cost black-box student models (e.g., GPT-4o mini). Given the costs of frontier API models (e.g., GPT-5) today, training with frontier student models would be beyond our limited research budget; training in a domain with 200 examples for 20 epochs and GRPO group size 8 would make for a total of $200 \times 20 \times 8 = 32,000$ calls to the frontier model, which we estimate to cost thousands of dollars. The modularity of ADVISOR MODELS allows us to train with lower-cost student models and transfer the learned advisor to frontier students, reducing costs by orders of magnitude.

### 4.1. ADVISOR MODELS outperform other optimizers in improving black-box models

On three domains requiring specialized reasoning likely to be sparse in frontier models' training corpus, we show that ADVISOR MODELS outperforms the state-of-the-art prompt optimizer GEPA (Agrawal et al., 2026) at improving black-box models. We then show in Section 4.2 how these advisors transfer the lifts they provide to frontier models.

**RuleArena Taxes** (Zhou et al., 2025) provides models with an individual's tax profile and the complete tax filing instructions and asks for a liability/refund value. Due to the complexity of the task, we use the Level 0 (easiest) subset and the 3-step ADVISOR MODELS architecture. Reward is 0/1 for incorrect/correct calculations. We sample 75 train and 25 test examples out of the available 100 examples.

**SWE Agent Efficiency** examines the behavior of ADVI-

SOR MODELS in a multi-turn domain. While standard SWE agent training focuses on accuracy, an underexplored and less saturated aspect of agent training is the efficiency with which it is able to solve tasks (# of interactions with the environment). We utilize mini-SWE-agent (Yang et al., 2024b) as the agent harness and a subset of SWE-smith (Yang et al., 2026), a dataset of curated SWE tasks for various Python repositories. We split 149 issues in a repository into 100 training and 49 test issues. Leveraging ideas from the curriculum learning literature (Narvekar et al., 2020; Bae et al., 2026), we filter the train split to remove difficult problems outside of the student model's capabilities, leaving a high learning signal train set of 56. We design a reward function accounting for both correctness and efficiency, and train with the multi-turn architecture injecting advisor generations every 5 student steps. Further details are provided in Appendix C.1.

**MTOB** (Machine Translation with One Book; Tanzer et al. (2024)) involves translating the extremely low-resource Kalamang language into English, requiring linguistic reasoning as well as specialized knowledge about Kalamang. Models are provided a source text and translations of substrings retrieved via string similarity and tasked with producing a translation. Following the original work, we use chrF (Popović, 2015), a measure of string similarity, between the generated translation and ground-truth translation as reward. We sample 200 training and 50 test examples and train with the 2-step architecture.

**Models and Baselines.** We train advisor models on Qwen models (Qwen Team, 2024; 2025) with student models from the GPT and Gemini families (OpenAI, 2024; 2025a; Gemini Team, 2025a). Advisor/Student configurations for each domain are in Figure 3, and were selected for capability/cost considerations (e.g., stronger models for RuleArena Taxes, Gemini models for SWE Agent Efficiency[1]). Training details are in Appendix A. We further provide ablations with Llama models as advisors in Appendix D.2.

To measure improvement, we compare the performance of our ADVISOR MODELS-trained pipelines to the pipeline before training (i.e., untrained advisor) and the student on its own. Additionally, we baseline against GEPA (Agrawal et al., 2026), a state-of-the-art static prompt optimizer that iteratively self-reflects over sampled traces. For fair comparison, GEPA was allowed to access the reward function an equivalent number of times as in ADVISOR MODELS training. The GEPA baseline was not run for SWE Agent Efficiency as the method is not optimized for multi-turn tasks and there's a high cost associated with training on the domain. Instead, for this domain only, we compare against

---

[1]We had access to Gemini credits when it came time to run the most expensive multi-turn SWE Agent Efficiency domain.

directly prompting the student-only baseline agent to be more efficient.

### 4.1.1. RESULTS

The performance of our trained ADVISOR MODELS pipelines (i.e., with the student models used during training) is shown in Figure 3. On SWE Agent Efficiency, Gemini 2.5 Flash on its own achieved a resolve rate of 61.2% requiring on average 19.1 interaction steps. Modifying the agent prompt to encourage efficiency resulted in statistically indistinguishable resolve rates and average interaction steps. In contrast, an untrained efficiency-aware advisor significantly improved the efficiency (13.7 steps), but at a significant cost to resolve rate (52.6%). However, after training, the full ADVISOR MODELS pipeline maintained the original resolve rate (61.2%) while retaining the significant efficiency gains (14.4 steps).

Inspecting advice at the start and end of training ADVISOR MODELS for the SWE Agent Efficiency domain, we found that an interpretable improvement the advisor learned was to provide immediately relevant advice and encourage the efficient exploration of file-systems via `grep` instead of `ls`. We provide further details of this qualitative example in Appendix E.1.

On MTOB, both the standalone student (GPT-4o mini) and untrained advisor baselines achieved similar chrF scores of 33.1 and 33.2 respectively. GEPA was unable to improve upon these baselines (33.0). The trained advisor provides significant improvements, lifting performance to 45.8, approaching the expert human performance of 51.6.

In RuleArena Taxes, GPT-4.1 mini alone achieved an accuracy of 64.8%. Introducing an untrained advisor, however, dropped the performance to 52.0%, indicating that a low-quality revision prompt can harm performance, in line with previous work on the harms of overthinking in LLMs (Cuadron et al., 2025). GEPA was able to mitigate this risk, improving over the untrained advisor (58.4%), but was not able to regain the standalone student baseline. On the other hand, the trained advisor overcomes the loss and improves upon the standalone baseline, achieving an accuracy of 76.8%.

### 4.2. ADVISOR MODELS lift the capabilities of frontier black-box models

On the same specialized reasoning domains, we show that the advisors trained with low-cost students can be transferred to improve frontier performance. We transfer to frontier models within the same family as the original student, i.e., GPT-5.2 (OpenAI, 2025c) for RuleArena Taxes, GPT-5 (OpenAI, 2025b) for MTOB and Gemini 3 Pro (Gemini Team, 2025b) for SWE Agent Efficiency. The results of the

these experiments are presented in Figure 4.

On SWE Agent Efficiency, the trained advisor with the original student (Gemini 2.5 Flash) sped up trajectories by 4.7 fewer average steps without affecting resolve rate. When replacing the student with the frontier Gemini 3 Pro, the trained advisor similarly sped up trajectories by 5.4 fewer average steps without affecting resolve rate. These results indicate that the advice the advisor learns to generate is interpretable and insightful, allowing a frontier model uninvolved in the training process to improve.

On MTOB, the gain provided by the trained advisor to GPT-5 is smaller in magnitude, but statistically significant (47.8 vs. 46.2, $p \approx 0.03$). This can partially be attributed to the limited headroom of improvement for GPT-5 on MTOB, where it already approaches the human expert baseline of 51.6. In this case, while the advice the advisor learns to generate was able to help fill in knowledge unknown by GPT-4o mini, GPT-5 has filled in most of the knowledge gap on its own, and the improvement is marginal.

On RuleArena Taxes, the advisor model trained with GPT-4.1 mini was able to improve GPT-5.2's performance from 67.2% to 85.6%. Across all three domains, ADVISOR MODELS-trained advisors can **improve the performance of frontier models**. Notably, these improvements were achieved at a reasonable cost without needing to interact with the more expensive frontier model directly during training. We ablate transferring to frontier models in different families as well in Appendix D.1.

We emphasize that the frontier model improvements were achieved by transferring advisors trained using a **low-cost** student to demonstrate the practicality of our method. We expect that training directly with frontier models would be even more effective.

### 4.3. ADVISOR MODELS can learn hidden preferences

On three personalization domains requiring learning undisclosed individual preferences purely through scalar rewards, we show that ADVISOR MODELS outperform static prompt optimizers and effectively personalizes black-box frontier models, similar to the reasoning domains. All preference domains use the 2-step architecture mentioned in Section 3.

**Review Writing.** We use a review-writing setup to test whether advisors can learn user-specific preferences in open-ended generation tasks. The system must produce media reviews for named users, with prompts drawn from the FSPO dataset (Singh et al., 2026). The 500 FSPO prompts are split into 450 training and 50 evaluation and equally distributed between 5 users. Each user is associated with a latent preference not stated in the prompt. We further study scaling the number of users to 100 in Appendix D.3

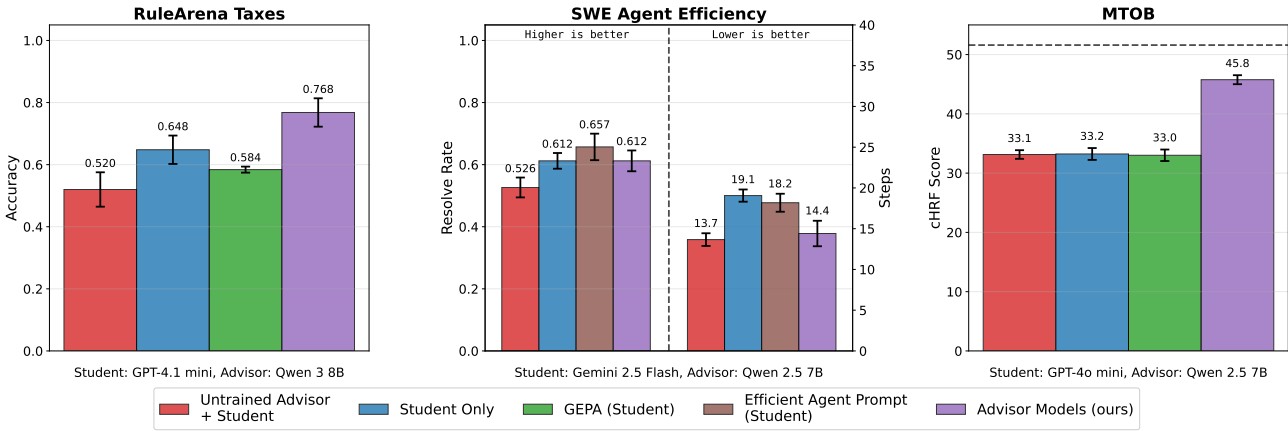

*Figure 3.* **ADVISOR MODELS improves over baselines in specialized reasoning tasks.** After training, ADVISOR MODELS can improve at tasks beyond the performance of the student model alone, though using an untrained advisor can hurt performance. The human performance on MTOB of 51.6 shown in dashed line here and when relevant in subsequent plots. 95% confidence intervals are provided.

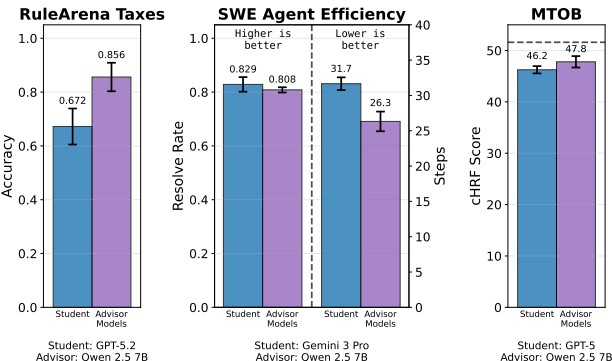

*Figure 4.* **ADVISOR MODELS improves frontier model capabilities.** Advisor models trained using low-cost students can be used to improve the performance of frontier models. 95% confidence intervals are provided.

We study two variants of preferences over the FSPO dataset. In **Review Length**, each user has a preferred length between 10 and 1000 words. The reward is designed to be 1 when the length exactly matches the preference and decay towards 0 as deviation grows; see Appendix C.2 for details. In **Review Level**, users have a preferred reading levels (e.g., elementary school, high school, college professor, etc). Reward is binary, using GPT-5 mini (OpenAI, 2025b) as a judge to determine whether the review matches the specified level.

**Math Solutions.** We also introduce the Math Solutions domain, where the task is to produce teaching material for individual students in the form of worked solutions to math problems sourced from the MATH-500 benchmark (Lightman et al., 2024). Each student either prefers or dislikes questions posed to the reader and either prefers or dislikes presentation of multiple methods, making for 4 total distinct preferences. We split the MATH-500 benchmark into 400 training and 100 evaluation problems. GPT-4.1 mini is used as a judge to determine whether or not a criteria is met, with

0 reward given if no preference is met, 0.4 if only one is met, and 1 if both preferences are met.

Compared to the review writing domains, Math Solutions requires the advisor to learn multiple axes of preference and allows us to verify that correctness of solutions is not degraded by ADVISOR MODELS training in Section 4.4.2.

**Models and Baselines.** For all personalization domains, we use GPT-4o mini as the student model and Qwen2.5 7B Instruct as the advisor model. Further training details are again in Appendix A. In addition to the baselines evaluated in the previous subsection, we compare against Profile-Augmented Generation (PAG; Richardson et al. (2023)), a personalization-specific static prompt optimizer that generates a preference profile for each user based on sampled traces. We also add an in-context oracle, where the exact preferences of all individuals are provided in the prompt.

### 4.3.1. RESULTS

We present our results on the three personalization domains in Figure 5. Across all three tasks, PAG and GEPA achieve performances that are marginally above the un-optimized baselines, if at all. To ensure the correctness of the baseline, we contacted the GEPA authors and worked with them to confirm a fair implementation of GEPA for these tasks.

On the other hand, ADVISOR MODELS achieves impressive results, essentially perfectly learning the preferences with 0.94 reward out of a maximum 1.0 for review length, 99.6% accuracy for review level, and 0.948 reward out of a maximum 1.0 for math solutions. This demonstrates an ability for ADVISOR MODELS to effectively learn in personalization settings where current static optimizers cannot.

Notably, ADVISOR MODELS even exceeds the performance of the in-context oracle. Whereas the in-context oracle

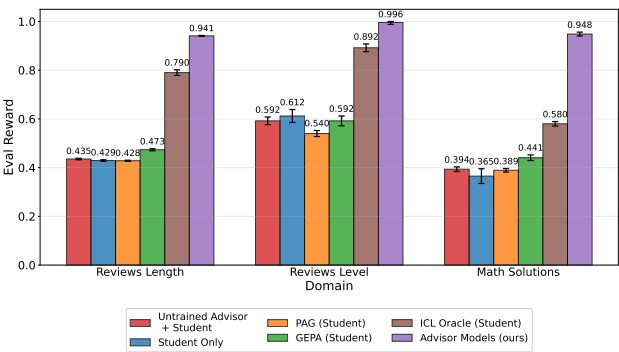

*Figure 5.* **ADVISOR MODELS effectively learns hidden preferences in personalization tasks.** ADVISOR MODELS can almost perfectly learn individual preferences, even outperforming static prompts with full information. Static prompt optimization methods provide at best marginal improvements. For all personalization domains, the student is GPT-4o mini and the advisor base is Qwen2.5 7B Instruct. 95% confidence intervals are provided.

prompts require the student model to identify the relevant preference and adhere to it, ADVISOR MODELS provide dynamic instance-relevant advice in a manner tailored to elicit the desired output from the student model. Examples of generated advice from the start and end of training in the review length setting are presented in Appendix E.2. Qualitatively, early advisor outputs hallucinate user preferences, but by the end of training they converge toward the true latent. These findings provide further evidence that dynamic, instance-specific advice allows black-box models to be effectively optimized in settings where static prompt optimization struggle.

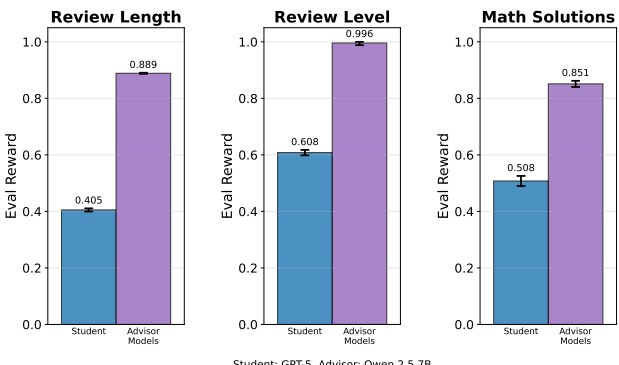

*Figure 6.* **ADVISOR MODELS trained on personalization transfer to frontier models.** Advisor models trained using low-cost students can be used to personalize frontier models. 95% confidence intervals are provided.

Figure 6 presents our results transferring the trained advisors to GPT-5. For all domains the trained advisor was able to personalize GPT-5 towards the individuals, indicating the interpretability of the generated advice and the ability of ADVISOR MODELS to personalize frontier models. Performance on Review Length and Math Solutions is not as

perfect as the results with the original GPT-4o mini student, which is because the advisor's generations are especially tailored to elicit desired responses out of GPT-4o mini.

### 4.4. Ablations

#### 4.4.1. ADVISOR PROMPT INITIALIZATION

No matter the specific design, prompt templates to elicit advisor and student generations are necessary to initialize ADVISOR MODELS. The template to the advisor includes the input as well as any additional priors that might help guide the optimization process, if available.

The choice of initialization prompt is less crucial for reasoning tasks, but for personalization tasks they can speed up training by guiding the model towards the desired axes of optimization. In our experiments, the initializations we used were relatively strong, including guidance to help the model sample relevant advice (e.g., "consider the length of the review"). We believe this to be reasonable as practitioners will have priors or candidates for axes of separation. Nevertheless, we investigate the efficacy of ADVISOR MODELS under weak initialization, when the advisor provides advice with no guidance. We train ADVISOR MODELS on the Review Length domain with a weak initialization prompt. The prompts (as well as the student prompt) are presented in Appendix F.

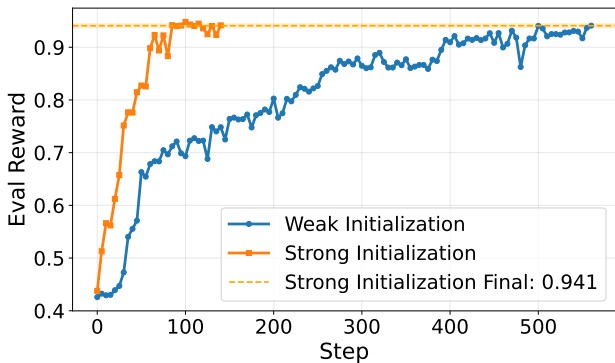

*Figure 7.* **Weak initialization learns just as well but slower.** ADVISOR MODELS learning curve on the review length domain with strong and weak initialization for 5 and 20 epochs, respectively. Final strong initialization performance provided as reference.

We find that under weak initialization, ADVISOR MODELS learn slower, but can still achieve the same level of performance as strong initialization with extended training (Figure 7). After 5 epochs strong initialization has fully learned the task (0.941) while weak initialization is still learning (0.749), but by 20 epochs weak initialization also attains full learning (0.941). This indicates that ADVISOR MODELS can learn hidden latents relying solely on the advisor's ability to sample into learning signal, but the process can be greatly accelerated by guiding the advisor. We be-

lieve that in practice, it is reasonable to assume some idea of the target axes, and so training with strong initialization prompts is not unreasonable, if not the norm.

### 4.4.2. CAPABILITY RETENTION

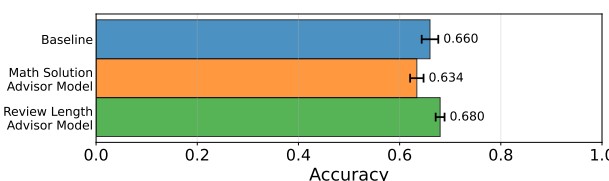

*Figure 8.* **Specializing an advisor does not degrade the student's core capabilities.** ADVISOR MODELS pipelines (Qwen2.5 7B Advisor & GPT-4o mini Student) trained on unrelated tasks show no degradation in accuracy on MATH-500 compared to GPT-4o-mini, demonstrating the robustness of ADVISOR MODELS. 95% confidence intervals are provided.

Another benefit of the modularity of ADVISOR MODELS is its inherent robustness against catastrophic forgetting. Unlike direct fine-tuning, which can degrade a model's general capabilities (Luo et al., 2025; Zhu et al., 2024; Zhang et al., 2023), our approach preserves the student model's weights, leaving its core competencies unaltered.

Figure 8 demonstrates this robustness. We take the AD-VISOR MODELS pipelines trained on Math Solutions and Review Length (from Section 4.3) and show their correctness on the MATH-500 dataset (Lightman et al., 2024) is essentially the same as the unadvised baseline. This indicates that ADVISOR MODELS training does not affect performance on un-optimized objectives, even when there is strong contextual overlap (as is the case for Math Solutions and accuracy on MATH-500). We provide further experiments highlighting the robustness of our system with frontier models in Appendix D.5.

### 4.4.3. COMPARISON TO RL

When access to the student model's weights is available, we expect direct reinforcement learning to provide greater gains than ADVISOR MODELS. While we study the setting where students are black-box, we ask how much of the gain of RL could be recovered by ADVISOR MODELS. To investigate, we conduct a training run where Qwen2.5 3B Instruct advises Qwen2.5 7B Instruct on the MTOB domain and compare against directly training Qwen2.5 7B Instruct. We train both methods using the hyperparameters described in Appendix 1 for 20 epochs, matching the setup used in Section 4.1.

The results (Figure 9) shows that both RL and ADVISOR MODELS make major improvements over the baseline chrF of 31.17 with 42.38 and 38.98, respectively. Further discussion and training details are in Appendix D.4. Though direct RL remains the better option, the ADVISOR MOD-

ELS pipeline is able to make up most of the improvement over baseline, indicating its strength as an alternative when student model weights are not available.

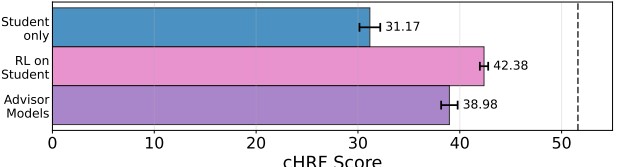

*Figure 9.* **ADVISOR MODELS vs. direct RL of student model on MTOB.** Direct RL remains the gold standard, but ADVISOR MODELS can account for most of the improvement over baseline (Qwen2.5 3B Advisor & 7B Student). 95% confidence intervals are provided.

### 4.4.4. LIMITS OF ADVISOR MODELS

A requirement for ADVISOR MODELS to be effective is that there must be *something* for the advisor to learn. For in-distribution domains where frontier models are already highly knowledgeable, there may be nothing the advisor can learn that the student doesn't already know. To demonstrate this, we trained and evaluated ADVISOR MODELS with a Qwen2.5 7B Instruct advisor and GPT-4o mini student on one such domain: math problem solving, specifically MATH-500. We use the same hyperparameters as other domains (Appendix A) and train for 20 epochs. Figure 10 shows that ADVISOR MODELS provide no significant improvement over the baselines as there is nothing the advisor can learn that would improve frontier capabilities at math.

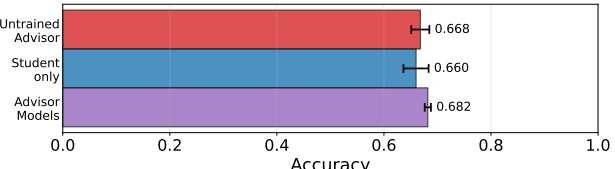

*Figure 10.* **ADVISOR MODELS on MATH-500.** As a highly targeted domain, frontier models are "maxed-out" on math problem solving, and so there is nothing an advisor learns to improve frontier models (Qwen2.5 7B Advisor & GPT-4o mini Student). 95% confidence intervals are provided.

## 5. Conclusion & Future Work

We introduced ADVISOR MODELS, a method for steering black-box foundation models through learned dynamic advice. Unlike static prompt optimization, advisors generate instance-specific guidance, enabling personalization and specialized reasoning beyond base model capabilities.

Our experiments demonstrate substantial abilities to lift frontier black-box models including 24.6% faster SWE agents, 27.4% improvement in complex rule-following, and near-perfect learning of hidden user preferences where static methods fail entirely. Critically, advisors trained on afford-

able models transfer improvements to frontier systems and preserve general capabilities.

ADVISOR MODELS addresses a growing need: as frontier models increasingly become black-box services, researchers and practitioners need methods to customize and improve them without weight access. Our method provides a practical, cost-effective solution that achieves specialization while maintaining robustness.

## Acknowledgements

We would like to thank Tyler Griggs and Lakshya A Agrawal for insightful discussions about this work.

Sky Computing Lab is supported by gifts from Accenture, Amazon, AMD, Anyscale, Broadcom, Google, IBM, Intel, Intesa Sanpaolo, Lambda, Lightspeed, Mibura, NVIDIA, Samsung SDS, and SAP. We would additionally like to acknowledge Databricks for their generous compute support.

This material is based upon work supported by the National Science Foundation under Grant No. DGE 2146752 and NSF IFML, CCF 2019844. Any opinions, findings, and conclusions or recommendations expressed in this material are those of the author(s) and do not necessarily reflect the views of the National Science Foundation.

## Impact Statement

This paper presents work whose goal is to advance the field of Machine Learning. There are many potential societal consequences of our work, none which we feel must be specifically highlighted here.

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

# A. Training Details

We utilize a lightly modified fork of the SkyRL framework for all of our RL training runs (Griggs et al., 2025). We run all training experiments unless otherwise noted on a single node of 8xH100s. For reproducibility, we provide key non-default training parameters used in Table 1; SWE Agent Efficiency used different hyperparameters to allow for the longer multi-turn rollouts. The number of training epochs used in each domain is shown in Table 2.

*Table 1.* Key training hyperparameters used for ADVISOR MODELS experiments. SWE Agent Efficiency has different values to support the longer multi-turn rollouts.

| HYPERPARAMETER | VALUE | SWE AGENT EFFICIENCY VALUE |
|---|---|---|
| TRAIN batch size | 16 | 8 |
| POLICY MINI-BATCH SIZE | 4 | 4 |
| MICRO-BATCH SIZE (PER GPU) | 2 | 1 |
| LEARNING RATE | $1.0 \times 10^{-6}$ | $1.0 \times 10^{-6}$ |
| MAX PROMPT LENGTH | 8192 | 28672 |
| MAX GENERATION LENGTH | 16384 | 4096 |
| TEMPERATURE | 1.0 | 1.0 |

*Table 2.* Training epochs for ADVISOR MODELS experiments. Note that SWE Agent Efficiency and RuleArena taxes have relatively small train sets due to small source datasets, hence the larger number of training epochs.

| DOMAIN | TRAINING EPOCHS |
|---|---|
| SWE AGENT EFFICIENCY | 30 |
| MTOB | 20 |
| RULEARENA TAXES | 40 |
| REVIEW LENGTH | 5 |
| REVIEW LEVEL | 5 |
| MATH SOLUTIONS | 10 |

# B. ADVISOR MODELS Variants

In this section we provide further discussion on variations upon the core ADVISOR MODELS framework, including variations not used in this paper.

## B.1. 3-step Advisor System

The 3-step architecture modifies the basic ADVISOR MODELS formulation to reduce the difficulty of the advisor's role. Rather than ask the advisor to provide advice given only the task prompt, it is also given an initial attempt by the student. There are thus 3 steps in this formulation rather than 2: first, the student model makes an initial attempt at the task; second, the advisor provides advice based on the attempt; third, the student model updates its attempt based on the advice.

Under this formulation, the advisor's role is made simpler by the fact there is an initial attempt to evaluate. This shifts the role of the advisor from a pure generator to a verifier. We found that this change allowed ADVISOR MODELS to learn more effectively in the RuleArena setting when compared to the standard 2-step formulation. While this does come at the cost of twice as many calls to the student model during inference, initial student generations in the train set may be obtained offline and kept static throughout the training process.

## B.2. Multi-Turn Advising

In multi-turn settings, the advisor has the opportunity to provide multiple pieces of advice throughout a single rollout. From the advisor's perspective, after it generates advice, an environmental step occurs, and the trajectory either terminates or the advisor receives an observation and generates advice again. The student is to use the advice to perform actions during the environmental step, and the system will track the student's actions and outcomes to build the observation for the advisor.

In the simplest setup, the student will take only a single action during the step. However, there is nothing restricting the

student from taking more actions. Indeed, in our implementation for SWE Agent Efficiency the student takes 5 actions before finishing its step. The system then collects all 5 actions taken by the student and their outcomes as the input for the advisor. Setting higher intervals can lead to faster training as fewer advisor calls are needed, but also reduces the impact of the advisor as advice becomes sparser in rollouts. One might also need to consider options such as summarization or truncation of student actions and observations to support lower frequency interactions.

### B.2.1. ADVISOR AS TOOL CALL

An alternative is to allow for a dynamic interval. Under this formulation, the student model would be able to choose when to end the step. From the student's perspective, it has the ability to pause the trajectory and request guidance from an advisor. From the advisor's perspective, every environmental step can contain a variable number of student actions. In theory, this should allow for the most efficient use of advisor generations as students ought only end a step when it is stuck and would benefit most from advice. In practice, however, we found that without being able to apply optimization pressure to the student model, frontier models are loathe to ask for help and the advisor was rarely called.

### B.3. Monte Carlo Reward Estimation

Here we describe an alternative method to calculate reward, rather than another ADVISOR MODELS architecture. The reward we calculate should be the reward assigned to the advisor's generations, i.e., a measure of how good the generated advice is. In the original ADVISOR MODELS formulation, we estimate the quality of the advice by measuring whether the student model is able to produce a good final output based on the advice. However, student model generations are non-deterministic, meaning this reward estimate has high variance.

One way to reduce this variance is through Monte Carlo reward estimation (Kazemnejad et al., 2025), where for a single piece of advice we take the average reward of $N$ student responses rather the reward of a single student response. This has the effect of giving a lower variance estimate of the reward for the advice. In theory, this should result in more stable training due to better credit attribution. In practice, this was prohibitively expensive for us as it multiplies the number of student model calls by $N$.

## C. Domain Specific Details

In this appendix, we provide further details about data and training setups for specific domains, especially those that use non-binary reward functions.

### C.1. SWE Agent Training

We source from problems built around the Tenacity library, chosen for its reasonable size and test cases to control training costs. The train set was filtered by evaluating each candidate problem 5 times using an unadvised Gemini 2.5 Flash mini-SWE-agent and removing all problems that did not have at least one successful resolve. Our reward function is designed such that the reward is 0 for unresolved issues and ranges from 0.5 and 1.0 as issues are resolved more rapidly, encouraging efficiency while retaining correctness. To control for cost, trajectories are capped at 40 environmental interactions from the student; trajectories that reach this cap are marked as unresolved.

$$\text{Reward} = \begin{cases} 0, & \text{if unresolved,} \\ 0.5 + 0.5 \cdot \dfrac{40 - \text{steps}}{40}, & \text{if resolved.} \end{cases}$$

### C.2. Review Length

The reward function is designed to provide reward of 1 when the length preference as matched exactly and decay towards 0 as the magnitude of the difference grows.

$$\text{Reward} = \frac{1}{1 + \frac{|\text{Review Length} - \text{Preferred Length}|}{\text{Preferred Length}}}.$$

# D. Additional Ablations

## D.1. Transfer Across Student Families

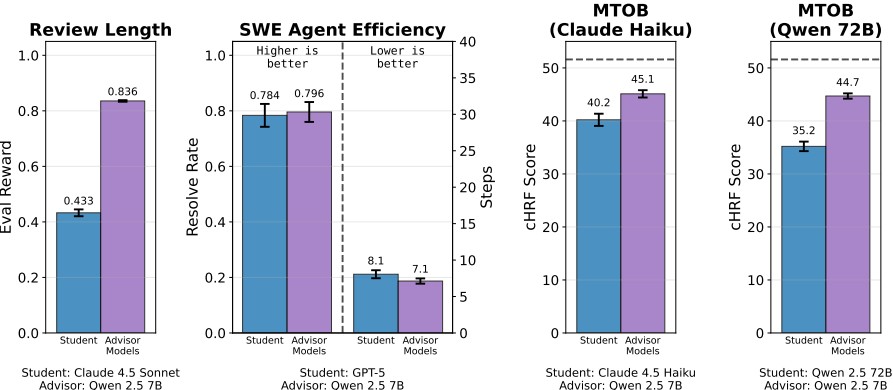

*Figure 11.* **ADVISOR MODELS-trained advisors transfer across model providers.** Advisor models trained using students from one model provider work well with students from other providers, demonstrating the interpretability of generated advice. 95% confidence intervals are provided.

The experiments in Section 4.2 show that advisors trained on low-cost models transfer effectively to a frontier model from the same model provider (e.g., GPT-4o mini to GPT-5).

Figure 11 shows the results of transferring advisors across model families. Our SWE Agent Efficiency advisor trained with a Gemini 2.5 Flash student continues to provide good guidance for GPT-5, achieving the same resolve rate as the GPT-5 standalone baseline (79.6% vs 78.4%) while reducing the average number of steps by 1 (7.1 from 8.1). As GPT-5 is already quite efficient there is less room for improvement, but the improvement is nevertheless statistically significant ($p \approx 0.02$).

On Review Length, the advisor trained on a GPT-4o mini student transfers to a Claude 4.5 Sonnet (Anthropic, 2025b) student by improving the average reward from 0.433 to 0.836. On MTOB, we transfer the advisor trained on a GPT-4o mini student to Claude 4.5 Haiku (Anthropic, 2025a), providing a chrF improvement from 40.2 to 45.1. Though Claude and GPT models are trained on different data mixtures, to further address the point on multilinguality, we performed an experiment on MTOB where we transferred the advisors that were trained with a student that is likely primarily English (GPT-4o-mini) onto a larger Qwen student (Qwen-2.5-72B-Instruct) that likely has higher Chinese content. **We continue to show strong transfer results, indicating that advisors can transfer well across students with different training distributions.**

Naturally, a trained advisor provides the greatest gains for the student model it was trained with as the learned advice is especially tailored to the original student. Still, the fact that new students, including from other model families, also exhibit improvement provides further evidence that the advisor learns to generate useful and interpretable natural language advice.

## D.2. Advisor Models w/ Llama

Figure 12 presents our results training ADVISOR MODELS with Llama 3.1 8B Instruct (Llama Team, 2024) advising GPT-4o mini on MTOB to show ADVISOR MODELS generalize to other advisor model families. We used the same training hyperparameters as described in Appendix A. Here, we see that the ADVISOR MODELS trained pipeline attains a performance of 42.52 chrF, a significant improvement over the GPT-4o mini baseline of 33.23 and the untrained pipeline baseline of 32.60. These results provide evidence that the utility of ADVISOR MODELS is not restricted to the Qwen family.

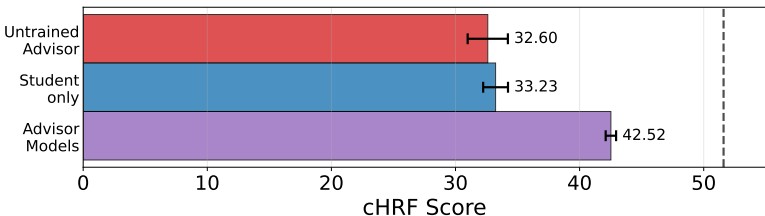

*Figure 12.* **ADVISOR MODELS with a Llama 3.1 8B Instruct advisor model on MTOB.** ADVISOR MODELS can also train Llama models to be effective advisors, demonstrating our method generalizes to different model families. 95% confidence intervals are provided.

## D.3. Scaling Personalization

In this ablation, we scale up the number of individuals in the Review Length domain to show ADVISOR MODELS can handle larger-scale personalization tasks. Each individual is randomly assigned a length preference between 10 and 1000 words. We sample 10 training and 5 test examples from each individual for a total of 1000 training and 500 test examples. As in the original Review Length domain (Section 4.3), we train a Qwen2.5 7B Instruct advisor for a GPT-4o mini student. We follow the same training setup as for the original Review Length domain (Appendix A), except we train for 10 epochs instead of 5. Due to the fewer number of training examples per individual, this actually results in $4.5\times$ fewer training rollouts per individual than the original Review Length domain.

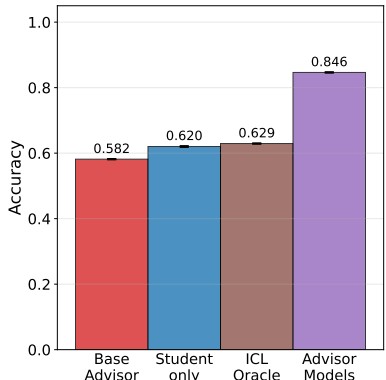

*Figure 13.* **ADVISOR MODELS on Review Length, scaled to 100 individuals.** With 100 individuals, even oracle static prompts are unable to help with personalization. The ADVISOR MODELS pipeline provides dynamic, instance-specific advice which effectively personalizes the student generations. 95% confidence intervals are provided.

Figure 13 presents our results. The performance of the student model alone was 0.62. We found that the in-context oracle prompt was unable to improve (0.63), demonstrating the difficulty of communicating all preferences in a single static prompt. On the other hand, ADVISOR MODELS training provides meaningful improvement, attaining 0.85 reward. Though less than the final performance of the original Review Length domain, the system has meaningfully personalized and has the potential to further improve as training reaches parity with the original in terms of rollouts per individual.

## D.4. Direct RL vs. ADVISOR MODELS

In this ablation, we ask how close ADVISOR MODELS come to replicating the effects of RL. Though RL is impossible on black-box models, we may conduct studies comparing ADVISOR MODELS using an open-source student model and RL on that student model. In keeping with the use of weaker advisors advising stronger students, we train a Qwen2.5 3B Instruct advisor with Qwen2.5 7B Instruct as a student on the MTOB domain, comparing the results against directly training Qwen2.5 7B Instruct on MTOB. For both methods, we use the Appendix A training details.

Results and discussion thereof are in Section 4.4.3; here we focus on the eval curves (Figure 14). We see that direct RL achieves fast learning early before flattening out, while the ADVISOR MODELS system steadily improves throughout. Notably, scores for both methods are still climbing at the end of training, indicating the potential for further improvement.

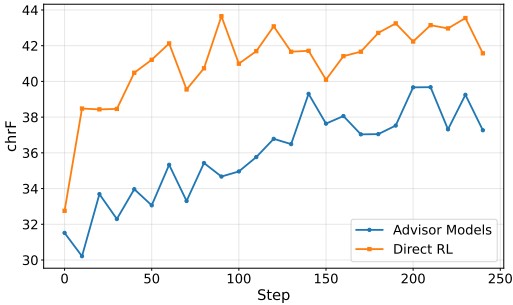

*Figure 14.* **Reward curves of ADVISOR MODELS vs. direct RL of student model on MTOB.** Direct RL performs better, but ADVISOR MODELS can provide a majority of the gains given by direct RL.

## D.5. Robustness of Frontier Models

In Section 4.4.2, we provided experiments highlighting how by keeping an unchanged black-box model, our proposed system is robust to handling inputs that may be out-of-distribution or from different domains. We chose to run robustness experiments on the weaker students used during training as we expect frontier models to be less sensitive to irrelevant advice. Weaker students may over-index on the irrelevant advice and make mistakes, whereas we expect frontier models to correctly ignore the irrelevant advice and achieve their original performance.

To further confirm this, we provide experiments demonstrating robustness of frontier models on our existing benchmarks, demonstrating the lack of degradation in capabilities even with an advisor trained on an unrelated task (review length) across a variety of domains in Figure 15.

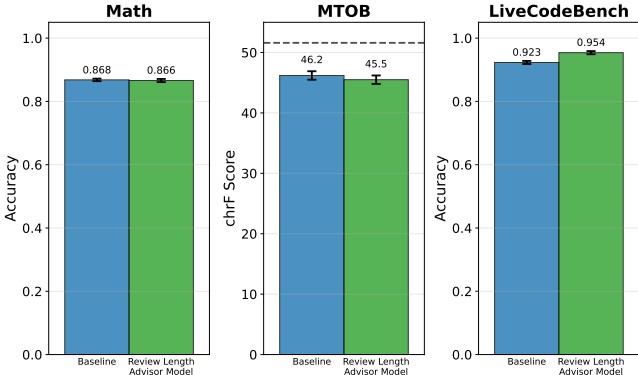

*Figure 15.* **ADVISOR MODELS robustness holds for frontier models as well.** ADVISOR MODELS pipelines (Qwen2.5 7B Advisor & GPT-5 mini Student) trained on unrelated tasks show no degradation in performance on MATH-500, MTOB, and LiveCodeBench Exec compared to GPT-5 alone, further demonstrating the robustness of ADVISOR MODELS. 95% confidence intervals are provided.

# E. Advice Examples

## E.1. SWE Agent Efficiency

In this appendix, we present examples advice generated by ADVISOR MODELS before and after training for the same evaluation problems in the SWE Agent Efficiency setting.

---

**SWE Agent Efficiency Eval Example**

**Issue Info**

**Statistics property returns unexpected data after recent changes**
**Description**
The statistics property on retry decorators is returning unexpected data that includes an "initial" key with value True, instead of the expected empty dictionary or proper statistics data.
**Steps/Code to Reproduce**
[OMITTED FOR SPACE]

**Advice at Step 0**

To quickly resolve the issue, follow these steps:

1. **Locate the `statistics` property implementation**

   - Navigate to the `tenacity` module where the `Retry` and `RetryCall` classes are defined.
   - Look for the `statistics` property in these classes.

2. **Inspect the `statistics` property**

   - Check whether an `__init__` method initializes the `statistics` attribute.
   - Verify that `statistics` is not left in a partially initialized state.

3. **Modify the `statistics` initialization**

   - Ensure the attribute is initialized explicitly, e.g., `self.statistics = {}` or `self.statistics = None`.

4. **Test the fix**

   - Run the reproduction scripts to confirm that `statistics` now returns an empty dictionary or valid retry statistics.

**Illustrative example fix:**
[CODE OMITTED FOR SPACE]
**Verification:**
[CODE OMITTED FOR SPACE]
If the `statistics` attribute is being set to `{'initial': True}` elsewhere in the codebase, ensure that logic is corrected or removed to prevent misleading state.

**Advice after Training**

First, inspect the `statistics` property implementation. Run:

```
grep -nIR 'statistics' *
```

This will help locate where `statistics` is defined and invoked within the codebase.

---

### E.2. Review Length

In this appendix, we present examples advice generated by ADVISOR MODELS before and after training for the same evaluation problems in the Review Length setting.

---

**Review Length Eval Example #1**

Prompt Info

**Name:** Matei
**Latent Preference:** 10-word long reviews
**Prompt:** Write a review for the Horror book, Frankenstein.

Advice at Step 0

Given Matei's preference for detailed analysis and a blend of personal critique and factual information, advise focusing on the book's complex themes, character development, and intriguing narrative structure. Matei benefits from a mix of objective observations and subjective experiences, so include both your personal reactions and insights backed by textual evidence. Since Matei enjoys detailed reviews, ensure your review is around 500-600 words, exploring the book's historical context, its impact on the genre, and your specific responses to the eerie atmosphere and psychological depth.

Advice after Training

Given Matei's preference for concise, straightforward reviews, focus on a brief, no-nonsense review. Keep your review to about 8-10 words, highlighting whether you enjoyed the book or not and why, if necessary. For example: "Frankenstein is a well-crafted horror book that is easy to read."

---

**Review Length Eval Example #2**

Prompt Info

**Name:** Alex
**Latent Preference:** 1000-word long reviews
**Prompt:** Write a review for the Drama movie, Titanic.

Advice at Step 0

For Alex, consider focusing on the emotional depth and historical significance of the film in your 3-4 sentence review. Since Alex appreciates detailed storytelling, mention how James Cameron masterfully combines romance with historical drama to create a moving narrative. Additionally, Alex might enjoy a touch on the visual and audio effects that enhance the Titanic's scale and the hardships faced by the characters.

Advice after Training

Given Alex's preference for detailed analysis, focus on discussing the plot, character development, and themes of the movie. Since Alex enjoys longer, more in-depth reviews, write a 1000-word review that explores not just the plot, but also the film's impact and its relevance today.

# F. Prompt Examples

In this appendix, we present the advisor and student prompts we used to train and evaluate ADVISOR MODELS on the Review Length domain.

---

**Review Length Advisor Prompt (strong)**

System Prompt

You are a review writing advisor. Provide specific guidance for writing a review that matches the person's preferences. Consider the length preferences and style that would work best for the target person.

Instruction

You need to provide advice for writing a review for {person}.
The task is: {prompt}
Provide specific advice about the review that would work best for {person}. Think carefully about the length of the review in your advice. Keep your advice to 3-4 sentences.

---

**Review Length Advisor Prompt (weak)**

System Prompt

You are a review writing advisor. Provide specific guidance for writing a review.

Instruction

You need to provide advice for writing a review for {person}.
The task is: {prompt}
Provide specific advice about the review for {person}. Keep your advice to 3-4 sentences.

---

**Review Length Student Prompt**

System Prompt

You are a review writer. Based on the prompt and advisor guidance, write a review that follows the guidance provided. Write a clear, well-structured review.

Instruction

Review Prompt: {prompt}
Advisor Guidance:
{advice}
Write a review following the advisor's guidance.

