# OpenReview forum: "How to Train Your Advisor: Steering Black-Box LLMs with Advisor Models"
_ICML.cc/2026/Conference — ICML 2026 regular_

### Official Review · Reviewer_i2RG · 2026-03-11

**Soundness:** 3
**Presentation:** 4
**Significance:** 3
**Originality:** 2
**Overall Recommendation:** 5
**Confidence:** 3

**Summary:**

Authors propose a method that trains a small prompt prefix generator (advisor model) that improves a target black-box model’s performance in certain tasks. Unlike previous works with LLMs that try to learn a single common prompt for all tasks or prompts, this work finds a per-instance prompt without the need for any labels.

The advisor model has several variants:
* 2-step pipeline: advisor generates a prompt prefix, student uses the prefix, get reward to upgate the advisor model using RL.
* 3-step or even multi-step pipeline: student attempts, advisor uses the attempt to generates prompt prefix, student then uses the prefix to generate a better attempt, and so on.

The authors show the advisor model’s  improvements in several tasks:
* reasoning tasks where domain knowledge is sparse in the pretrained model
* Better performance on a tax task and machine translation tasks.
* Less degradation in performance in coding, but more efficient with shorter COT
* Preferences that cannot be known a priori.

Show that advisor models trained to optimize smaller black-box models can be transferred to larger black-box models.

**Compliance With Llm Reviewing Policy:**

Affirmed.

**Key Questions For Authors:**

1) The idea of a dynamic instead of static prompt for prompt optimization is not a novel idea. Prior works has also tried to learn a small input-conditioned prompt model for a frozen target LLM without any labels (see [1] as an example), with the major difference that this work was done before the LLMs era and hence employed different training strategies. Authors should cite the relevant works appropriately and clarify this idea itself is not a new contribution of the paper.

[1] IDPG: An Instance-Dependent Prompt Generation Method, NAACL 2022,  https://arxiv.org/pdf/2204.04497

2) How did the authors specifically select the level 0 tax task in Rule Arena? Have authors also tried Airline and/or NBA tasks in Rule Arena, and other levels in the tax task?

3) The authors showed only one task (swe) to demonstrate efficiency gains. Have authors shown this on any other longer horizon tasks like theorem proving?

4) Do the authors have any intuition about which tasks Advisor models would improve in efficiency versus performance?


5) > For fair comparison, GEPA was allowed to access the reward function an equivalent number of times as in ADVISOR MODELS training.

Is this a fair comparison since GEPA is training-free (backward passes are typically 2x FLOPa than  forward pass). Are the advisor models compute-matched with the baselines?

6) > Interestingly, on RuleArena Taxes, GPT-5 performs worse than GPT-4.1 mini

Do we know how much worse? Do authors know if this is due to prompt template sensitivity? I.e. have authors tried other prompt templates and still see similar performance?

7) For section 4’s results section, the authors do not stick to a single student or teacher model, and rather switches between them, is this choice completely random, or were these models chosen intentionally?

8) Regarding Robustness: Authors make a general claim that the advisor model can be transferred from a smaller to a larger target LLM. Is this true for any models, even if they were pre-trained on very different data mixtures? I.e. If you train with Qwen models as the target, which are suspected to be trained on more Chinese tokens, and hence are more efficient in Chinese .  When you evaluate on Chinese math tasks. Do you still expect to observe the same efficiency gains when applied to a non-Chinese model, such as GPT5?

**Limitations:**

Yes

**Strengths And Weaknesses:**

Strength:

* Simple idea with well-executed experiments.

* The paper is easy to follow.

* Method yields interpretable advices.

Weaknesses:

* See questions below

---

> ### Author Rebuttal · Authors · 2026-03-31
>
> We thank the reviewer for their constructive feedback. We appreciate the recognition of the
> **“well-executed experiments”**. Below, we clarify several questions and address suggestions for
> additional analysis through additional experiments that further strengthen the results.
>
> - Q6: GPT-5 on RuleArena Taxes
>   - As mentioned in the paper, we are also unsure why GPT 5 is worse at RuleArena
> Taxes than GPT 4.1 mini, but we note it's not unheard of for “stronger” models to
> be worse at a particular task. We did run our transferability experiments with the
> newer GPT 5.2 model that does yield better baseline performance than GPT 4.1
> mini. This provides additional evidence that the GPT 5 performance drop is an
> anomaly. Our advisor is **still able to lift performance**, further strengthening our
> transferability claims.
>
>   | Setting | Accuracy |
>   |-|-|
>   | GPT-5.2 no advisor | 67.2% (± 6.7%) |
>   | GPT-5.2 + trained advisor | 85.6% (± 5.3%) |
>
> - Q8: Transfer Across Different Pre-Training Mixes
>   - While we did transfer across model families, we didn’t explicitly perform
> experiments across countries of origin. In Appendix D, we describe how we took
> advisors that were trained for GPT students and transferred them to Claude
> student models, and advisors that were trained for Gemini students and
> transferred them to GPT models. We transfer across the Review Length, SWE
> Agent Efficiency, and MTOB domains, consistently showing gains.
>
>   - Though Claude and GPT models are trained on different data mixtures, to further address the point on multilinguality, we performed an experiment on MTOB where we transferred the advisors that were trained with a student that is likely primarily English (GPT-4o-mini) onto a larger Qwen student (Qwen-2.5-72B-Instruct) that likely has higher Chinese content, as the reviewer mentions. We **continue to show strong transfer results, indicating that advisors can transfer well across students with different training distributions**.
>
>   | Student Model | No Advisor | w/ Trained Advisor |
>   |-|-|-|
>   | GPT-4o-mini | 0.332 ± 0.010 | 0.458 ± 0.008 |
>   | Claude 4.5 Haiku | 0.402 ± 0.012 | 0.451 ± 0.007 |
>   | Qwen 2.5 72B | 0.352 ± 0.009 | 0.447 ± 0.005 |
>
> - Q5: Comparison to GEPA
>   - The GEPA paper itself actually uses reward function call count to compare
> against RL methods, and we followed GEPA’s matching method. Further, given
> the performance of GEPA we saw, we do not believe GEPA would make any
> further improvements even with more FLOPs, and we consulted with the authors
> of GEPA to ensure that we were running a fair baseline (bottom of page 6, RHS).
>
> - Q1: Related Work
>   - We thank the reviewer for bringing this work to our attention and will discuss and
> cite it accordingly in a camera ready version!
>
> - Q2: Rule Arena
>   - We selected RuleArena Taxes level 0 as a domain upon which we could rapidly
> and cheaply test our method. Harder difficulties would be more difficult to obtain
> a learning signal for the advisor/student combination. We did not investigate the
> other domains as we wanted to have a wide spread of domains in total and only
> had limited compute; Taxes level 0 was selected out of the RuleArena options as
> the easiest subset of a challenging domain.
>
> - Q7: Choice of models
>   - We switched between models to show the general applicability of our method
> across multiple model families and capabilities. Our goal was not to produce a
> single method for a single base model, but rather to show the method is a
> generally applicable training method. As mentioned in the paper, cost was also a
> consideration in this choice; we used Gemini for the most expensive SWE Agent
> Efficiency run as we had access to Gemini credits.
>
> - Q3: Efficiency
>   - The multi-turn SWE Agent Efficiency task was the only domain where we
> examined agent efficiency. Due to compute and cost constraints we could only
> conduct experiments on one long-horizon multi-step domain. However, applying
> Advisor Models to other long-horizon multi-step domains such as theorem
> proving is an exciting direction for future work.
>
> - Q4: Efficiency vs performance
>   - We believe that given enough compute it may be possible for Advisor Models to
> improve in performance of SWE resolve rate, but a challenge is that current
> frontier models are already quite over-tuned for the SWE resolution task, and so
> advice injection may not make much difference. As mentioned in the paper, we
> believe that Advisor Models excel best at specialized tasks the student is not
> over-tuned on. For out-of-domain long-horizon tasks, we would expect to see
> improvements in performance as well, though due to compute constraints we are
> not able to verify this.

---

> > ### Author Rebuttal · Reviewer_i2RG · 2026-04-02
> >
> > Dear authors,
> >
> > Thank you for your rebuttal and your efforts in running additional experiments. My concerns have been addressed, and I will maintain my rating.

---

### Official Review · Reviewer_JHmq · 2026-03-11

**Soundness:** 3
**Presentation:** 3
**Significance:** 3
**Originality:** 3
**Overall Recommendation:** 4
**Confidence:** 3

**Summary:**

This paper proposes a method for training small and open-weight language models to generate dynamic natural language advice that steers frozen black-box LLMs. The Advisor is trained using GRPO on small models with low-cost student models and then transferred to frontier students without further tuning.

**Compliance With Llm Reviewing Policy:**

Affirmed.

**Final Justification:**

I thank the authors for their response. The rebuttal addresses some of my questions and clarifies several previously ambiguous aspects of the paper, although some concerns are still not fully addressed. I have also considered the other reviewers’ comments and the follow-up discussion with the authors. Taking all information into account, I keep my rating unchanged, and my final recommendation remains Weak Accept.

**Key Questions For Authors:**

Please refer to the Weaknesses.

**Limitations:**

Yes

**Strengths And Weaknesses:**

Strengths:
-	The core idea is clear which reframes prompt engineering as an RL problem where the advisor policy learns which advice has better performance from the interaction between a frozen student and an environment.
-	The transferability result is a good contribution which advisors trained with cheap student models transfer improvements to frontier models without further tuning.

Weaknesses:
-	The paper reports that GPT-5 performs worse than GPT-4.1 mini on RuleArena Taxes, which is anomaly, and the advisor’s improvement of GPT-5 could be partly attributed to correcting an unusually low baseline rather than demonstrating general frontier model improvement. This needs to be investigated.
-	The capability retention task relies solely on MATH-500 with GPT-4o mini as the student, which is a weak proxy for robustness. The frontier models would be near or at ceiling on MATH-500, making degradation effectively undetectable even if it occurred. A convincing robustness test should use a diverse set of benchmarks spanning clearly distinct domains, e.g., coding, medical QA, etc., and should be run directly on the frontier model students used in the transfer experiments.
-	Given the small training sets, sensitivity to training set size and variance across training seeds is not studied. For example, how much variance is there across independent training runs?

---

> ### Author Rebuttal · Authors · 2026-03-31
>
> We thank the reviewer for their constructive feedback. We appreciate the recognition of the
> novelty of the contributions behind the work. Below, we clarify several questions and **run all
> suggested experiments to address all concerns** and further strengthen the results. Please let
> us know if there is anything else that would be convincing in recommending this paper more
> strongly.
>
>
> - W3: Training Seeds
>   - We agree with the reviewer that running additional experiments on more seeds
> would further strengthen our results, and we run four additional experiments with
> new seeds on the MTOB domain. Our results below show that all four of the new
> **training runs with different seeds showed similar improvements over the
> baselines**, highlighting the robustness of Advisor Models training:
>
>   | Setting | chrF |
>   |-|-|
>   | Student only | 33.23 (± 1.00)|
>   | GEPA | 33.01 (± 0.97)|
>   | w/ Untrained Advisor | 33.14 (± 0.73) |
>   | Advisor Models (Paper Result) | 45.76 (± 0.76) |
>   | Advisor Models (Alt. Seed 1) | 44.13 (± 0.65) |
>   | Advisor Models (Alt. Seed 2) | 42.73 (± 0.26) |
>   | Advisor Models (Alt. Seed 3) | 40.21 (± 0.28) |
>   | Advisor Models (Alt. Seed 4) | 42.55 (± 0.36) |
>
>   - Regarding dataset size, we used all data available for RuleArena (100 points split into 75 train and 25 test) and used the full eval set of size 50 for MTOB. For SWE Agent Efficiency, filtering was necessary due to compute limitations; we required a high-signal train set to best use the limited compute available to us. For all results we provide error bars.
>
> - W2: Robustness
>   - We chose to run robustness experiments on the weaker students used during
> training precisely because we expect frontier models to be less sensitive to
> irrelevant advice. Weaker students may over-index on the irrelevant advice and
> make mistakes, whereas we expect frontier models to correctly ignore the
> irrelevant advice and achieve their original performance.
>
>   - Nevertheless, as the reviewer asked, we have run the experiments demonstrating robustness of frontier models on our existing benchmarks, **demonstrating the lack of degradation in capabilities** even with an advisor trained on an unrelated task (review length) across a variety of domains:
>
>   | Domain | GPT-5 Alone | GPT-5 + Review Advisor (OOD) |
>   |-|-|-|
>   | Math | 86.8% (± 0.4%) | 86.6% (± 0.5%) |
>   | MTOB | 46.2% chrF (± 0.7%) | 45.5% chrF (± 0.7%) |
>   | LiveCodeBench Exec | 92.3% (± 0.5%) | 95.4% (± 0.5%) |
>
> - W1: GPT-5 on taxes
>   - As we note in the paper, we are also unsure why GPT 5 is worse at RuleArena
> Taxes than GPT 4.1 mini, but we note it's not unheard of for “stronger” models to
> be worse at a particular task. We did run our transferability experiments with the
> newer GPT 5.2 model that does yield better baseline performance than GPT 4.1
> mini. This provides additional evidence that the GPT 5 performance drop is an
> anomaly. Our advisor is **still able to lift performance**, further strengthening our
> transferability claims.
>
>   | Setting | Accuracy |
>   |-|-|
>   | GPT-5.2 no advisor | 67.2% (± 6.7%) |
>   | GPT-5.2 + trained advisor | 85.6% (± 5.3%) |
>
> Given the additional results provided for all suggested experiments that further strengthen the
> paper and clarification to questions, we ask **if you’d consider recommending this paper as a
> stronger accept or if there is any further information that would be convincing?**

---

> > ### Author Rebuttal · Reviewer_JHmq · 2026-04-01
> >
> > Thank you for the authors’ rebuttal. The rebuttal provides additional details that help clarify and address some of the questions I raised. Though there are still some concerns, based on my previous comments and the authors’ response, I will maintain my weak accept rating.

---

### Official Review · Reviewer_au7n · 2026-03-11

**Soundness:** 2
**Presentation:** 2
**Significance:** 3
**Originality:** 3
**Overall Recommendation:** 3
**Confidence:** 4

**Summary:**

The authors propose Advisor Models  to improve the performance of black‑box frontier Large Language Models from GPT and Gemini families by training a small open‑weight advisor model to generate per‑instance natural‑language guidance that is injected into the black‑box model’s prompt.Advisor Models solve this by reframing prompt engineering as reinforcement‑learned policy optimization: the advisor observes the task, generates tailored advice, the black‑box model produces an output based on that advice, and the advisor is updated using rewards derived from the final outcome.

Across three diverse tasks, the authors show that advisor models outperform state‑of‑the‑art static prompt optimization methods. The advisor model  improve GPT‑5’s accuracy on the RuleArena Taxes benchmark from 31.2% to 53.6%; in multi‑turn software‑engineering environments, they reduce Gemini 3 Pro’s step count from 31.7 to 26.3 and for personalization tasks they raise reward scores from 40–60% to over 85–100%. The authors demonstrate  transferability through training the advisors using inexpensive models such as GPT‑4o mini and then deploying on frontier models like GPT‑5.

**Compliance With Llm Reviewing Policy:**

Affirmed.

**Key Questions For Authors:**

1. How sensitive is the advisor’s learning process to sparse, delayed, or noisy rewards from the black‑box model?
2. Is there a quantitative cost–benefit analysis comparing advisor‑augmented inference to baseline student-only inference?
3. Does the added prompt length disproportionately affect models with smaller context windows?

**Limitations:**

yes.

**Strengths And Weaknesses:**

Strengths:
1. The paper makes an original contribution by reframing prompt optimization as learning a trainable policy that generates per‑instance natural‑language advice.
2. Experimental analysis is supported by evaluations across multiple domains.

Weaknesses:

1. The advisor is optimized only through the reward derived from the student (black-box) model output, it is not guaranteed to learn optimal strategies for all instances.
2. Although advisor training is done on small models, during inference the advisor must produce advice for every instance and the student must consume longer prompts (input + advice) which leads to increased  latency and token usage.

---

> ### Author Rebuttal · Authors · 2026-03-31
>
> We thank the reviewer for their feedback. We appreciate the recognition of our strong empirical
> results and **“original contribution”**. Below, we clarify several questions and **run all suggested
> experiments** for additional analysis that further strengthen the paper. Please let us know if
> there is any other information that would be convincing to raise your score.
>
>
> - Q1: Sensitivity to sparse, delayed, or noisy rewards
>   - To explicitly test the concern around noisy rewards, **we performed an
> experiment on the MTOB domain where we added Gaussian noise** sampled
> from N(0, 0.05) to the reward during training; reward is based off of chrF which
> ranges from 0 to 1, though reward typically range from 0.3 to 0.45 throughout
> training. Here we found that the **Advisor Models training was robust to the
> noise**, achieving a performance that far surpassed all baseline methods and
> comparable to the baseline Advisor Models training with noiseless rewards
> across various training seeds (further experiments we ran):
>
>   | Setting | chrF |
>   |-|-|
>   | Student only | 33.23 (± 1.00)|
>   | GEPA | 33.01 (± 0.97)|
>   | w/ Untrained Advisor | 33.14 (± 0.73) |
>   | Advisor Models (Paper Result) | 45.76 (± 0.76) |
>   | Advisor Models (Alt. Seed 1) | 44.13 (± 0.65) |
>   | Advisor Models (Alt. Seed 2) | 42.73 (± 0.26) |
>   | Advisor Models (Alt. Seed 3) | 40.21 (± 0.28) |
>   | Advisor Models (Alt. Seed 4) | 42.55 (± 0.36) |
>   | Advisor Models (with Noisy Reward) | 40.42 (± 1.02) |
>
>   - Our experiments do explicitly test for sparse and delayed rewards by construction of the tasks, and we found that the Advisor Models method was very robust. For example, the SWE Agent Efficiency task is an example of both extremely sparse and delayed rewards as the agent may take up to 40 steps (unique generations) before it gets a single score at the very end that applies to all of its generations.
>
> - W1: Optimal instance-specific strategies
>   - The advisor’s description of the learning mechanism is accurate, but we do not
> claim to learn optimal strategies for each and every instance. There are,
> however, differences between instances that cannot be captured by a static
> prompt and this is what Advisor Models can learn.
>
> - W2, Q2, Q3: Increased cost and prompt length
>   - While Advisor Models do require an extra model call at inference, spending more
> compute at inference time is now a common strategy to improve performance.
> We may view advisor models as a learned chain-of-thought process that
> precedes further compute from the student model. Moreover, our results on the
> SWE Agent Efficiency setting shows that Advisor Models can guide API models
> to solve problems more rapidly, resulting in an overall efficiency increase.
>
>   - With respect to prompt length, we believe that Advisor Models are best used in conjunction with API models (otherwise direct RL on the student model would suffice). At the context lengths API models operate on, context length is not a limitation for most tasks.
>
>   - In terms of cost-benefit analysis, amortizing the cost of training to 0, we are able to obtain significant gains at negligible additional cost in tokens, just requiring the serving of a small 7/8B model. We believe for many users the gain in performance is worth the slight additional cost.
>
> Given the results we provide for all requested experiments, our clarifications for concerns that
> are not fundamental to the paper, and the reviewer's acknowledgment that the paper makes **"an
> original contribution" that is "supported by evaluations across multiple domains," would
> you consider raising your score** or letting us know what additional information would convince
> you?

---

> > ### Author Rebuttal · Reviewer_au7n · 2026-04-02
> >
> > Thank you for the responses, which have helped clarify some of the concerns. However, the justification for the need for advisor models remains unconvincing. I will maintain my original score.

---

> > > ### Author Response · Authors · 2026-04-03
> > >
> > > Thank you for noting that our response has clarified your original concerns. We provide further details on the motivation behind Advisor Models.
> > >
> > > As the reviewer notes, and we state in the paper, the purpose of Advisor Models is “to improve the performance of black‑box frontier Large Language Models” by “training a small open‑weight advisor model”. An important motivation for Advisor Models is that black-box frontier models are more capable than any open-weight model, but they are more difficult to optimize or specialize further. Optimization of black-box models by end users is restricted to brittle prompt modifications, whereas open-weight models can be directly trained.
> > >
> > > Our work is **essential** as the **only contemporary work** to provide a mechanism through which **reinforcement learning can be used to lift the performance of frontier black-box models**. We will emphasize this point in a camera-ready version.
> > >
> > > As we explain in Related Works, existing methods for customizing black-box models rely on static prompt optimization that create a single prompt to use for all instances of a task, but different instances may have different properties that different prompts would suit. Advisor Models parametrically learn to generate per-instance advice that is better able to improve model performance over all instances of a task. Our experiments demonstrate that learned models that generate per-instance advice **significantly improve frontier model performance beyond the lifts given by all state-of-the-art baselines**.

---

### Official Review · Reviewer_ye3o · 2026-03-13

**Soundness:** 3
**Presentation:** 3
**Significance:** 3
**Originality:** 3
**Overall Recommendation:** 3
**Confidence:** 2

**Summary:**

The paper proposes Advisor Models, a framework for improving black-box LLMs by training a small open-weight advisor model to generate instance-specific natural-language advice that is inserted into the black-box model’s context. The advisor is trained with GRPO using only task rewards from the black-box model’s final outputs, so the student model itself remains frozen. The paper studies both specialized reasoning and personalization, and emphasizes that advisors trained on cheap student models can transfer to stronger frontier models such as GPT-5 and Gemini 3 Pro.

**Compliance With Llm Reviewing Policy:**

Affirmed.

**Final Justification:**

I appreciate the authors’ rebuttal. The additional experiments and clarifications improved the paper in several meaningful ways: the extra seed runs on MTOB strengthen the stability story, the weak-initialization result clarifies the role of initialization in personalization, the GPT-5.2 transfer result makes the transferability claim more convincing, and the added cost discussion is useful. The rebuttal improved my confidence in the empirical results and in the authors’ care in addressing reviewer concerns.

That said, I am keeping my recommendation at 3 (Weak Reject). The paper has some strengths: the core idea is interesting and well motivated, the method is clearly presented, and the empirical improvements are promising. However, my main concern remains about scope and interpretation. Many of the key results are still supported by relatively small-scale, filtered, or controlled settings. RuleArena Taxes and MTOB are small, and the SWE Agent Efficiency training setup uses a filtered subset of issues. Likewise, the personalization experiments are well designed as controlled demonstrations, but they still fall short of establishing robustness in more realistic personalization settings with noisy, partial, or shifting user preferences.

I still think the paper’s mechanistic story is weaker than the empirical story. The rebuttal gives a reasonable intuition for why advice may transfer, but I remain unconvinced that the paper clearly distinguishes genuinely transferable advice from prompt-specific interference, or explains when transfer should or should not be expected.

**Key Questions For Authors:**

- How stable are the core results across seeds and alternative train/test splits, especially on the small datasets? RuleArena uses 25 test examples, MTOB uses 50, and SWE training is filtered down to 56 issues. More variance would increase the confidence in the results.
- How much of the personalization performance depends on strong initialization prompts that already name the relevant axis of preference?
Figure 7 suggests this matters substantially for sample efficiency
- How should we interpret the GPT-5 underperformance on RuleArena Taxes relative to GPT-4.1 mini?
- Can the authors provide a fuller cost breakdown, including GPU cost, API cost, and inference-time overhead from advisor calls?

**Limitations:**

- The paper should discuss that many of the strongest experiments are small-scale or synthetic. That applies both to the specialized reasoning tasks with small train/test splits and to the personalization tasks, which use stylized latent preferences and LLM judge
- The paper should also discuss practical cost and deployment complexity more explicitly. Training still requires RL on a 7B/8B advisor, API calls to the student, and in the reported setup a single node of 8xH100s.

**Strengths And Weaknesses:**

### Strengths
- core idea of turning black-box prompting into a trainable policy problem rather than a static prompt search problem is interesting and well motivated
- student-model results on specialized reasoning are strong and task-specific
- transfer-to-frontier story is supported in the results. Advisors trained on cheaper students still help frontier models: on GPT-5, RuleArena Taxes improves from 31.2% to 53.6%; on Gemini 3 Pro, SWE average steps fall from 31.7 to 26.3 with resolve rate retained; and on GPT-5 for MTOB, chrF improves from 46.2 to 47.8.
- personalization results are unusually large relative to the baselines. In Figure 5, the trained advisor reaches 0.941 reward on Review Length, 0.996 on Review Level, and 0.948 on Math Solutions, whereas static baselines such as PAG and GEPA are only marginally above the unoptimized baselines
- paper is generally easy to follow architecture is well explained


### Weaknesses
- mechanism behind the gains is not really unpacked. The method is “advisor + GRPO + reward from student output,” but there is little deeper analysis of why advice transfers, when it should help, or what separates good advice from superficial prompt interference beyond a few qualitative examples
- Several datasets are small, and one major domain is filtered in a way that weakens the claim. RuleArena Taxes uses only 75 train / 25 test examples. MTOB uses 200 train / 50 test examples. For SWE Agent Efficiency, the original 100 training issues are filtered down to 56 by removing problems “outside of the student model’s capabilities,” which helps focus learning signal but also makes the benchmark easier and narrows the evaluation.
- Personalization tasks are only strong as controlled experiments, but they are still synthetic. Review Length assigns users target lengths between 10 and 1000 words; Review Level uses a GPT-5 mini judge for reading level; Math Solutions uses a GPT-4.1 mini judge for two latent teaching-style preferences; I would hesitate to make the leap to real world personalization for real users with noisy, partial, or shifting preferences.
- Figure 7 shows that weak initialization learns much more slowly which mean the main personalization setting is not pure latent discovery from scratch.

---

> ### Author Rebuttal · Authors · 2026-03-31
>
> We thank the reviewer for their constructive feedback. We appreciate the recognition of our **strong empirical results, clarity, and motivation** behind the work. Below, we clarify several questions and **provide results for all suggested experiments** that further strengthen the paper. Please let us know if there is any more information that would be convincing to raise your score.
>
> - W2, Q1, L1: Seeds and Data Size
>   - We agree with the reviewer that running additional experiments on more seeds would further strengthen our results, and we run four additional experiments with new seeds on the MTOB domain. Our results below show that all four of the new
> **training runs with different seeds showed similar improvements** over the baselines, highlighting the robustness of Advisor Models training:
>
>   | Setting | chrF |
>   |-|-|
>   | Student only | 33.23 (± 1.00)|
>   | GEPA | 33.01 (± 0.97)|
>   | w/ Untrained Advisor | 33.14 (± 0.73) |
>   | Advisor Models (Paper Result) | 45.76 (± 0.76) |
>   | Advisor Models (Alt. Seed 1) | 44.13 (± 0.65) |
>   | Advisor Models (Alt. Seed 2) | 42.73 (± 0.26) |
>   | Advisor Models (Alt. Seed 3) | 40.21 (± 0.28) |
>   | Advisor Models (Alt. Seed 4) | 42.55 (± 0.36) |
>
>   - Regarding dataset size, we used all data available for RuleArena (100 points split into 75 train and 25 test) and used the full eval set of size 50 for MTOB. For SWE Agent Efficiency, filtering was necessary due to compute limitations; we required a high-signal train set to best use the limited compute available to us. For all results we provide error bars.
>
> - Q3: GPT-5 on taxes
>   - As we note in the paper, we are also unsure why GPT 5 is worse at RuleArena Taxes than GPT 4.1 mini, but we note it's not unheard of for “stronger” models to be worse at a particular task. We did run our transferability experiments with the newer GPT 5.2 model that does yield better baseline performance than GPT 4.1 mini. This provides additional evidence that the GPT 5 performance drop is an anomaly. Our advisor is **still able to lift performance**, further strengthening our transferability claims.
>
>   | Setting | Accuracy |
>   |-|-|
>   | GPT-5.2 no advisor | 67.2% (± 6.7%) |
>   | GPT-5.2 + trained advisor | 85.6% (± 5.3%) |
>
> - W4, Q2: Initialization
>   - As Figure 7 shows and the reviewer notes, strong initialization accelerated the learning process by conditioning the advisor to generate higher-signal advice. However, weak initialization was able to discover the latent eventually; both strong and weak initialization converge to the same final performance. Weak initialization is learning from “scratch”, though by the nature of Advisor Models the advisor must be initialized with some prompt to generate something.
>
> - Q4, L2: Cost Breakdown
>   - Most of our runs took less than a day to complete, with some of the personalization runs taking just a few hours. At current rates of just under ~\\$30/hr for 8xH100s, this would cost on the order of \\$100s for compute. The multi-turn SWE Agent Efficiency run took a few days, which would cost a few thousand dollars. The cost of API model calls during training was approximately an order of magnitude lower compared to compute costs.
>
>   - Additional inference latency depends on the compute used to serve the Advisor Model. We find that our trained 7B/8B advisors can be rapidly served using just a few H100s. Moreover, our results on the SWE Agent Efficiency task indicate that Advisor Models can actually reduce inference time latency by providing advice that speeds up API models.
>
> - W3: Personalization tasks
>   - We acknowledge that personalization, particularly with shifting preferences, is an open problem, and we do not claim to solve that problem, a point we will emphasize in a camera version ready of the paper. However, we believe our results demonstrate the ability of Advisor Models to discover environment latents purely through scalar reward signals **even when reward is obfuscated by the student model’s response**, as evidenced by its ability to discover static preferences otherwise not revealed to the advisor.
>
> - W1: Unpacking the Mechanisms
>   - Advisors must learn to generate natural language that improves the performance of the student model, and the crux of transferability lies in the observation that natural language is generally interpretable. There may be idiosyncrasies specific to the student model, but we hypothesize and demonstrate through our model transfer experiments that advice is helpful in general. More broadly, good advice is advice that improves model performance, and advice generated by our advisors does improve model performance.
>
> Given the results we provide for all requested experiments, our clarifications for concerns that are not fundamental to the paper, and the reviewer's acknowledgment that the **method is "well motivated" and the results are "strong", would you consider raising your score** or let us know what additional information would help?

---

> > ### Author Rebuttal · Reviewer_ye3o · 2026-04-04
> >
> > Thank you for the detailed rebuttal. The additional experiments are helpful. In particular, the extra seed runs on MTOB, the weak-initialization result, the added GPT-5.2 transfer result on Taxes, and the cost discussion all improve the paper and clarify several of my original questions.
> >
> > My main remaining concern is that the paper’s broad claims are still supported primarily by small-scale or filtered settings. RuleArena Taxes is very small, MTOB is also limited in size, and the SWE Agent Efficiency training set is filtered to a subset of issues that are within the student model’s capabilities. This does not invalidate the results, but it does make me hesitant about the strength and generality of the conclusions.
> >
> > I also still think the personalization evidence is mainly in controlled synthetic settings rather than realistic user personalization. The rebuttal appropriately acknowledges this for shifting preferences, but such a limitation remains important for interpreting the scope of the contribution.
> >
> > In addition, my mechanism concern is not fully resolved. The authors response gives a reasonable intuition for why advice may transfer, but I still do not think the paper clearly explains what separates genuinely useful transferable advice from prompt-specific interference, or when transfer should be expected to fail.
> >
> > Finally, the added cost discussion is useful, but it is still approximate rather than a full breakdown of training cost, API cost, and inference-time overhead.
> >
> > The rebuttal strengthens the existing results, but the remaining issues concern the core interpretation and scope of the results, and I am not certain they can be fully addressed within a short rebuttal. I am therefore keeping my score unchanged.

---

> > > ### Author Response · Authors · 2026-04-06
> > >
> > > We thank the reviewer for the response. We appreciate that the reviewer finds our claims are supported by the evidence. We believe that the remaining concerns can be addressed through minor writing fixes.
> > >
> > > Our dataset sizes are small, but the results show that Advisor Models can learn without the need for large amounts of data. For all our experiments, Advisor Models was able to significantly improve over all baselines on a held-out test set. We use all available data for RuleArena Taxes and the full test set for MTOB. For SWE Agent Efficiency, we filtered to a high-signal training set to manage costs for the expensive multi-step agent run, as is standard in RL (curriculum learning).
> > >
> > > For the personalization tasks, our primary intention is to demonstrate Advisor Models in a controlled setting with ground-truth advice for the Advisor Model to learn. We ensure the student model cannot possibly have an accurate prior on what to generate by synthesizing a personalization task. We demonstrated that Advisor Models learn to generate advice exactly in alignment with expected ground-truth advice in these settings. Realistic non-synthesized personalization was not feasible for us as we do not have ready access to large-scale user studies for online reward collection and offline evaluation. We will qualify our claims and make explicit our limitations in a camera-ready version of the paper, but don’t think they detract from the core contribution.
> > >
> > > We hint at the effect of prompt interference in Line 299 (left), explaining that we expect Advisor Models to provide the largest lifts for the student models they were trained with. The concern about learning prompt interference and not general advice is valid and worthy of discussion, however, overall, our transfer experiments show that Advisor Models excel at learning generally interpretable advice, evidenced by the transferable lifts across all experiments. We will include further discussion in a camera-ready version.
> > >
> > > As mentioned, exact costs depend heavily on the domain and types of compute/models used. For us, costs were on the order of a few hundred dollars per final training run. We cannot provide exact cost breakdowns as our bills aggregate costs across experiments and we have access to negotiated prices that we are unable to disclose. For replicability, all experiment details are included in the appendix.

---

### Decision · Program_Chairs · 2026-04-30

**Decision:**

Accept (regular)

**Comment:**

This paper proposes Advisor Models, a compelling framework that reframes prompt optimization for black-box LLMs as a trainable policy via reinforcement learning, enabling instance-level natural language guidance that consistently improves downstream performance. Reviewers agree the idea is well-motivated, clearly presented, and empirically promising, with strong gains across reasoning, agent efficiency, and controlled personalization, as well as evidence of transferability from weaker to frontier models. However, concerns remain about the scope and generality of the conclusions, as many results rely on small-scale, filtered, or synthetic settings, and the personalization experiments do not yet reflect realistic, noisy user preferences. Overall, the paper presents an interesting and potentially impactful direction, I am inclined toward a weak accept if there is still room.